# Co-production for the integration of migrant human capital into the decent work

**Valentina Gomes Haensel Schmitt** [1]*, **Agnieszka Ewa Olter-Castillo**[2], **Mirza Marvel Cequea**[3], **Helder Huaranga Chayña**[4]

1 Escuela de Negocios y Economía, Pontificia Universidad Católica de Valparaíso, Valparaíso, Chile,
2 Faculty of Geography and Regional Studies, University of Warsaw, Warsaw, Poland, 3 Escuela de
Ciencias Empresariales, Universidad Católica del Norte, Coquimbo, Chile, 4 Instituto de Desarrollo de la
Investigación Científica, Universidad de Lima, Santiago de Surco, Peru

* valentina.schmitt@pucv.cl

## Abstract

During the last few years, the mass exodus of Venezuelan citizens to other countries has turned Venezuela into an emigration country, with Peru receiving the second-highest number of migrants. This article explains the co-production process of integrating Venezuelan migrants and refugees into the host society under decent work conditions. This is a qualitative, cross-sectional explanatory case study covering the second half of 2022. It shows that within the co-production effort, civil society organisations entail a role in creating collective mechanisms and spaces to enable communication and information, helping to identify existing problems and vulnerabilities, and finding alternatives to mitigate them. Additionally, in the process of integrating migrants' human capital under the decent work condition there is the relevance of migrant-based civil society organisations, due to the fact that they can enhance trust and the quality of the interaction and dialogue with the beneficiary public, identify their specific needs and potential solutions and, therefore, increase the quality and efficiency of the services provided.

## 1. Introduction

Migration has been a part of human history from the earliest times [1]. It refers to the process of moving people from their place of residence within the territory of a country or to other countries [2]. It can be motivated by political, social, economic, or environmental factors [3] and may include people with a high level of education and professional experience, who leave their countries in search of economic and living conditions improvements, which is also called "human capital brain drain" [4–8]. However, forced migrants face additional vulnerabilities, which lead to increased risks and violation of their rights [9, 10]. Migration irregularity results in greater labour informality, as well as instability and abuse in working conditions [11].

The unstable economic and political situation in Venezuela over the last decades has led to a multidimensional crisis, which has been unprecedented in the country's history [12]. For almost two centuries, due to economic prosperity, that South American country had been an immigration country [13]. However, economic, political, and social changes [14, 15] triggered

**Data Availability Statement:** Sharing of individual
participant data with third parties was not
specifically included in the informed consent of the
study. Consequently, the information obtained

from interviews has been anonymized to safeguard the identities of the respondents. Due to ethical constraints regarding data sharing, custody of the data will be entrusted to a designated non-author point of contact, namely, the Scientific Research Institute of the Universidad de Lima. This institution will serve as the primary liaison for handling requests pertaining to data access. Researchers who wish to inquire about access to: idic@ulima.edu.pe To gain access, data requestors will need to sign a data access agreement and obtain the approval of the local ethics committee.

**Funding:** This work was financially supported by the Universidad de Lima (Peru) through the Instituto de Investigación Científica research grant number PI.52.001.2022. Fund was awarded to VGHS. The funder provided support in the form of salaries to VGHS from April 2022 to March 2023. All authors had full access to all the data in the study and accepted responsibility for submitting the publication. The specific roles of these authors are articulated in the 'author contributions' section. The funders had no role in study design, data collection and analysis, decision to publish, or preparation of the manuscript.

**Competing interests:** The authors have declared that no competing interests exist.

one of the largest displacement crises in the world [16] and the largest in the region in recent history [17]. Consequently, Venezuela has become a country of emigration [18]; and by June 2023, there were 7.32 million Venezuelan refugees and migrants in the world. Most of them were located in Latin America and the Caribbean, with Colombia, Peru, and Ecuador, being the three most relevant receiving countries in the region [19]. Peru is also the country with the highest indicators in refugee applications [20].

In Peru, most Venezuelans have settled in the capital [21], now representing 10% of the inhabitants of Metropolitan Lima, Peru [19]. In 2018, the presence of the Venezuelan population in Peru contributed 0.33 percentage points to GDP growth and accounted for 25% of private spending growth [21, 22]. In 2019, the Venezuelan population had a net positive fiscal impact [11]. However, despite the positive impact on the Peruvian economy, the living and working conditions of Venezuelans have often led to greater instability and vulnerability [23]. Also, since the beginning of migration, most Venezuelans have faced a high level of vulnerability, being unable to access public services [24].

During the COVID-19 pandemic, the vulnerabilities faced by migrants in terms of health became evident as they did not have access to the necessary services to cope with the crisis [25]. As of March 2020, to protect citizens, Peruvian government invoked a state of national emergency and mandatory social isolation came into effect in response to the pandemic outbreak [26]. The adopted measures resulted in unequal access to decent work [27], while informal labour rose to 76.1% in the period of June 2021-July 2022 [28]. In the case of Venezuelans, informality indicators reached about 90% [24], while less than 3% had been able to validate their professional degrees [29].

In this complex context of crisis, various actors have played a significant role in the support and integration process, resulting in a joint effort of co-production between citizens and public, private, and civil society organisations. As for the mass migration of Venezuelans in Peru, non-state actors seek to address poverty reduction, educational and labour inclusion, and the welfare of the most vulnerable, especially women and children [30].

This article explains the co-production process of integrating Venezuelan migrants and refugees into the host society under decent work conditions. The work is carried out from the perspective of human capital, migration, and decent work, focusing on the co-production efforts made by these organisations in Metropolitan Lima, Peru. The study presents the role, the interaction process, and the relevance of the different actors in the co-production process for integrating migrant and refugee population; thus, it seeks to ensure decent work, in line with the Sustainable Development Goals.

## 2. Literature review

### 2.1. Human capital and migration processes

Human capital refers to all the skills and capabilities that individuals can acquire over time to increase their productivity and income by improving their knowledge [31–33]. This capital serves as the foundation for economic growth in the various regions and countries where it has been applied and used correctly [34]. Therefore, it may be considered also a valuable resource for territorial and organizational development.

In migration processes, the adaptation of new human capital is intended to foster positive relations between migrants and local communities with the aim of ensuring the optimal development and utilisation of its potential [35]. At the local level, immigration not only impacts wages and taxes but also alters the composition of the local population [36] and influences the labour market dynamics [37]. This shift may potentially disrupt the socio-demographic balance that native inhabitants experience in their day-to-day lives [36, 37] and possibly raise

more concerns among individuals with lower educational attainment [36]. Moreover, the migration of individuals with lower skill levels may lead to modest competition between migrants and native inhabitants [38]. Integration can be understood as a ¨two-way¨ process of mutual accommodation of host communities and immigrants [39]. Socioeconomic integration of immigrants is considered one of the most important paths when seeking common prosperity, harmony, and social stability [40]. In this sense, migration movements might have a positive impact on social and economic welfare of the host country, as they can rejuvenate the working population and intensify the labour force. Therefore, it is essential that the host country provides support in terms of governance and protection of migrant workers [3].

Often, host countries do not recognise the professional skills and qualifications of migrants, which impedes the development of skills in migrant human capital [41]. Migrants are often positioned at the bottom of social pyramids with an increased vulnerability [42] and inserted into a highly informal labour structure [43]. Besides, discrimination against migrants can negatively affect the probability of accessing higher-income jobs, as their skills are less valued and rewarded [44, 45]. Although staying in the host country facilitates cultural integration over time and may increase migrants' education quality, it does not always neutralise the negative impact of the economic gap [46, 47]. Thus, providing the necessary tools for immigrants to develop their capacity to integrate into the labour market is important to contribute economically to the host country by obtaining new jobs and improving the perception that native people have of them [48]. The role of the host country becomes significant in the integration of new human capital [49], leading to "human capital gain" or "brain grain" [7].

## 2.2. Migration and decent work

Decent work refers to productive, well-paid work job opportunities with favourable conditions, security, and social protection [3, 50]. In terms of justice, emphasis is placed on four pillars of job creation, social protection, workers' rights and social dialogue [51, 52].

In migration process, it is crucial to prioritize efforts aimed at guaranteeing and promoting decent jobs, as well as strengthening governance and capabilities of national and local governments for achieving adequate socio-labour integration [3]. Moreover, even though employers may play a key role in that process, they often lack the necessary tools to ensure productivity in decent working conditions. Consequently, migrant workers' physical and mental health, and job satisfaction are negatively affected [53, 54].

The vulnerability of migrants and refugees is given by different factors, such as socioeconomic condition, gender, life cycle, race, ethnicity and territory, which subjects them to precarious jobs [9]. Given this reality, migrants who do not have the corresponding documentation are often more exposed to factors that violate their rights [55], such as labour exploitation or sexual harassment [3, 20]. Therefore, it is necessary to develop public–private initiatives that work towards achieving the Sustainable Development Goals [56]. Likewise, the creation of targeted programmes, as a result of collaboration of the main institutions and actors of the global economy is critical to ensure decent work conditions [3], and to reduce the multiple facets of vulnerability that might affect both the citizens of a country, the migrants and refugees [57].

## 2.3. Migration and co-production

According to Ostrom [58], co-production is a process in which the inputs provided by individuals who do not belong to a given organisation, are used and transformed into goods or services. It can also be understood as a model where there is the assumption of an active and participatory population [59–61] in the design, development, delivery of public services

[59, 60, 62] and in the execution and implementation of public policies, which ultimately improves the quality and efficiency of such services [60, 61, 63]. The combination of efforts by public service producers and citizens in the creation of goods and services can occur directly or indirectly [59, 64]. Co-production can also serve as a tool for ensuring the accountability of public officials by the population [65]. Consequently, the delivery of services relies on the cooperation of citizens and government agents [60, 66].

In the coproduction process, it is vital to share leadership to some extent in order it to be effective [67]; thus, a collective leadership approach may be necessary. Co-leadership arrangements involve multiple actors who jointly occupy a shared leadership role, with competing demands, different sources of expertise, and legitimacy. It allows for the development of smooth relations while also maintaining and mobilizing the tensions that can make collaboration most fruitful [68].

In migration processes, co-production seeks to provide public goods and services through collaboration with migrants' representatives as intermediaries within state agencies. In that way, an environment is created where migrants feel represented and at the same time, they manage to generate influence in terms of governance and representativeness within state bodies [69]. The work of non-governmental organisations within the framework of cooperation with the state is important, as it brings together different actors to discuss the challenges and possible solutions to achieve the integration into the host society [70].

The networks produced serve to enhance institutional links to build trust [71] and improve rights based migration governance [72, 73]. In crisis contexts, co-production is highlighted by an organisation's adherence in sharing responsibilities and power in the production and delivery of key public services. Likewise, support from institutional structures, business and political activities are important to generate more confidence and security for migrants [74].

## 3. Materials and methods

### 3.1. Methodology

This work presents a qualitative study, with explanatory case study characteristics [75], representing the organisations' co-production effort in integrating the Venezuelan migrants' human capital in the territory of Metropolitan Lima, Peru, under decent work conditions. The material constitutes a cross-sectional study, showing the existing reality during the second half of 2022. Yin [75] mentions that the single case study is justifiable under one of five conditions, among which a rare or exclusive circumstance, a typical or representative case, or revealing can be evidenced—so that the present case complies with the three described conditions.

Likewise, it is important to point out that the findings cannot be interpreted as a generalized representation with respect to the co-production processes of integrating migrants and refugees into the host society; However, it represents an important initial look into the phenomenon.

For secondary data, we analysed documents related to the subject, following the recommendation of Lee [76], published by governmental, multilateral, and civil society organisations. Primary data collection consisted of: 1. Semistructured interviews, such as proposed by Yin [77] with managers and representatives of public and private organisations, with an emphasis on civil society organisations, working on integrating human capital of Venezuelan migrants in Metropolitan Lima, Peru; 2. Non-participant observation was conducted, as described by Richardson [78], during the months of August and September 2022, for 50 days, when the study team attended migrant activities and workshops conducted by one of the organisations that supported the study.

For population representation, we considered the composition of a sample with a purpose as a criterion, with the adequacy to the parameters of the research questions, objectives and

purposes [79]. For this study, 82 organisations, participating in the Refugees for Venezuela (R4V) movement and the Cross-sectional Workgroup for Migration Management (MTIGM), were contacted by email and social networks. Among them 45 replied, stating that they were either available, unable to participate due to institutional regulations, or not suitable for the study. Ultimately, the sample comprised 22 representatives from 18 organisations.

## 3.2. Procedures

Firstly, the Ethics Committee of the Scientific Research Institute of the University of Lima, with opinion No. 027-CEI-IDIC-ULIMA-2022, has decided to approve the ethical viability of our research project. Then, participants were sent a standardised message inviting to take part in the study, and informed that the selection criterion was to be a managing member of an organisation involved in the integration process of Venezuelan migrants as productive subjects in Lima's labour market. Moreover, the participants were notified that their involvement would be voluntary and without cost, and that they could opt to terminate the interview at any time, if they wished to do so.

The interviews were conducted between October and November 2022, in face-to-face and virtual mode, depending on each participant's accessibility and preference. The average duration was 40 minutes, ranging from 30 minutes to 1 hour and 30 minutes. To ensure accurate documentation of the information, authorisation to record was requested. Also, it was explicitly stated that taking part in the interview posed no risk to the participants, and the interview content would be used only for this specific study and not be shared where else–as approved by the Ethics Committee. The study incorporated an informed consent procedure to provide participants with the necessary information to freely decide on their involvement in the research and the authorisation for using their related data. The consent was provided verbally. Finally, the identity of the participants has been treated with confidentiality, and the information they provided has been analysed collectively with the responses of the other participants.

The information collected during the expert interviews has been analysed through the content analysis technique [80], with the support of using NVivo 11 software. The interview script comprised 30 questions, which were finally interpreted by data triangulation [81], and arranged in sections: 1) Professional experience and projects related to the Venezuelan population; 2) General situation of the Venezuelan population in Metropolitan Lima; 3) Situation of the Venezuelan population in the labour market of Metropolitan Lima; 4) The co-production process, considering the role of the Peruvian State, civil society organisations and companies in the integration of the Venezuelan population into the labour market.

## 4. Results

### 4.1. Co-production in migration processes related to socioeconomic integration: The context

The co-production effort in migration processes related to socioeconomic integration is interdisciplinary, tackling matters concerning protection, health and nutrition, basic needs and education [82]. To address the mass migration of Venezuelans to Peru, among the different efforts, the Working Group for Refugees and Migrants (GTMR) has been created within the country, being composed by a joint platform, under a co-leadership model, made up of United Nations agencies, national and international non-governmental organisations (NGOs), international organisations, religious organisations, academia, embassies, donors and financial institutions. It provides space to coordinate and execute actions aiming at protection, assistance, and integration of refugees and migrants from Venezuela in Peru [82]. Its actions are

based on the fulfilment of coordination objectives, preparation and implementation of the Regional Response Plan for Refugees and Migrants [82]. This initiative requires coordination, communication and exchanging of experiences among organizations and its members to foster a more targeted response, where each sector contributes complementary actions.

The collaborative work between organisations seeks to reinforce and facilitate growth and planning, with the goal of establishing a clear and broad collective vision and objectives. Ideally, the connection between partners involved in the projects should enable different members to operate with a cohesive and unified vision, align and complement their efforts, and exert influence in the creation and implementation of public policies and organisational management practices. By engaging in the cross-sectional working groups enables organisations to accelerate their learning process by identifying diverse needs and collaborating across multiple sectors. Cross-sectional work seeks to enable the provision of safer and more secure access to essential services. Furthermore, organisational learning allows adopting actions that consider groups of migrants, refugees, and citizens of the host country with similar needs. A mixed approach helps to broaden the connection between different audiences, foster empathy, and integration, reduce biases and achieve care the goals by generating funding mechanisms based on resources from different origins. However, it is not always the reality, as in some cases organizations may understand each other as working under competing interests and resources, lack of communication and understanding and not necessarily generating an effective effort.

In the co-production process, a key feature presented is the scope of actions involved, which may be international, national, regional and/or local. The work is carried out jointly by organisations and representatives from various sectors such as the government, civil society and its organisations, international organisations, companies, religious organisations, academia, among others. The effort seeks to generate initiatives that improve living conditions, with the ultimate goal of ensuring harmonious, peaceful and inclusive migration process. Therefore, various societal actors unite their effort to achieve results with the resources available, including monetary, material, human and structural resources, forming an integrated, comprehensive and interdisciplinary group. Collaboration between organisations involves working at different levels, with a complementarity approach and a direct engagement with the community. Groups are created to undertake projects, exchange information, establish alliances and expand their outcomes. The idea is that collaboration among the different organisations to enable the achievement of their objectives.

Regarding the specific case of Venezuelans in Metropolitan Lima, Peru, efforts to facilitate their integration into the labour market include addressing legal aspects related to employability and the generation of enterprises. Legal barriers are the main limitation to their entry into the labour market, and the ability to secure the necessary means and quality of life. Also, within the realm of legal aspects, both basic and highly complex issues must be addressed, such as obtaining an immigration status and corresponding documentation, as well as the recognition of degrees and professional licensing when applicable. The primary obstacles and restrictions are characterised by legislation with decontextualised features, limited public access to information, public officials providing inadequate attention, lengthy and costly procedures. On the other hand, organisations focused on integration seek to provide information, guidance and financial support for those with professional degrees to obtain recognition from the State and the corresponding professional associations.

In this sense, the co-production efforts to enhance employability involve taking advantage of the human capital of migrants and refugees facilitating their integration into the labour market, recognising their potential and expanding their human capital. The training offered by the partner organisations includes specialised courses that cultivate skills needed in the local market, which aims to improve their chances of being hiring and integrated into the

workforce. The content encompasses job training services, self-employment training and certification of labour skills. Furthermore, there is a concerted effort to direct employability initiatives of migrants and refugees towards formal market insertion, and to reinforce basic aspects of employability, important for both the local and migrant populations. However, participation on it is surrounded by limitations as most of migrants under vulnerable situation need to dedicate its time to short-term needs and income generation.

Initiatives directed towards enterprises prioritise the need for a strategy to diversify the businesses of the refugee and migrant population, leveraging their skills and knowledge, allowing for scalability—and not just mere survival. The efforts focus also on identifying legal aspects currently in effect and those that require changes. Organisations work to generate information, analyse the current situation and engage, with communities to learn about their realities, understand their problems and create strategies accordingly. Therefore, in terms of training, the contents include managerial and legal aspects, topics relevant to the scope of sustainable development and actions aimed at promoting new competitive areas.

With respect to the labour integration of migrants and refugees, organizations seek to provide transversal activities of fundamental importance, such as providing services for strengthening emotional skills, taking the gender approach, enhancing public communication, and providing high-quality information. The work of strengthening emotional skills consists of offering psychological and psychiatric support for those who are affected by their vulnerable condition. The gender focus becomes especially relevant, as women and children are identified as the most vulnerable groups—followed by LGBTQIA+ groups and elderly people. The communication efforts aim to reduce conflicts between the migrant population and the local community, avoid xenophobic incidents, promote positive narratives and relationships, and broaden the potential for socio-productive integration. At the same time, it is important to offer objective and high-quality information to enable individuals to access, in a structured manner, livelihoods and improve their quality of life. Basic needs such as access to housing, health, education and ensuring food security have direct and indirect impacts on the processes of integration, so NGOs and international organisations collaborate closely to bridge the existing gaps, despite facing certain limitations in their actions.

The financial aspect is particularly relevant, as it serves both as motivator for migration and a limiting factor in the quality of life of migrants and refugees. Access to financial services is crucial for generating sustainable impact. Therefore, efforts are made to bring the ecosystem and allied financial entities closer to the actions taken, enabling migrants and refugees to access to these services. Financial inclusion is carried out by distinct organisations that aim to create mechanisms enabling migrants and refugees to access the financial system. These initiatives include facilitating monetary transfers, providing support for the payment of professional recognition processes, supplying seed capital for enterprises, and offering financial education.

Fig 1 shows how the co-production is generated, where it is designed with the migrants and refugees and delivers the service to the migrants and refugees.

## 4.2 The role of organisations in co-production

This broad and complex effort to foster complementarity and cooperation involves the actions of various institutions, emphasising the critical role of the State, private organisations, and civil society.

**4.2.1. The role of the state.**   In general, the State has the responsibility of facilitating co-production by reducing barriers, facilitating opportunities, and bridging development gaps at the macro and at the micro level. Also, by representing the State, the government assumes responsibility for public policy making and regulation. Public management involves designing

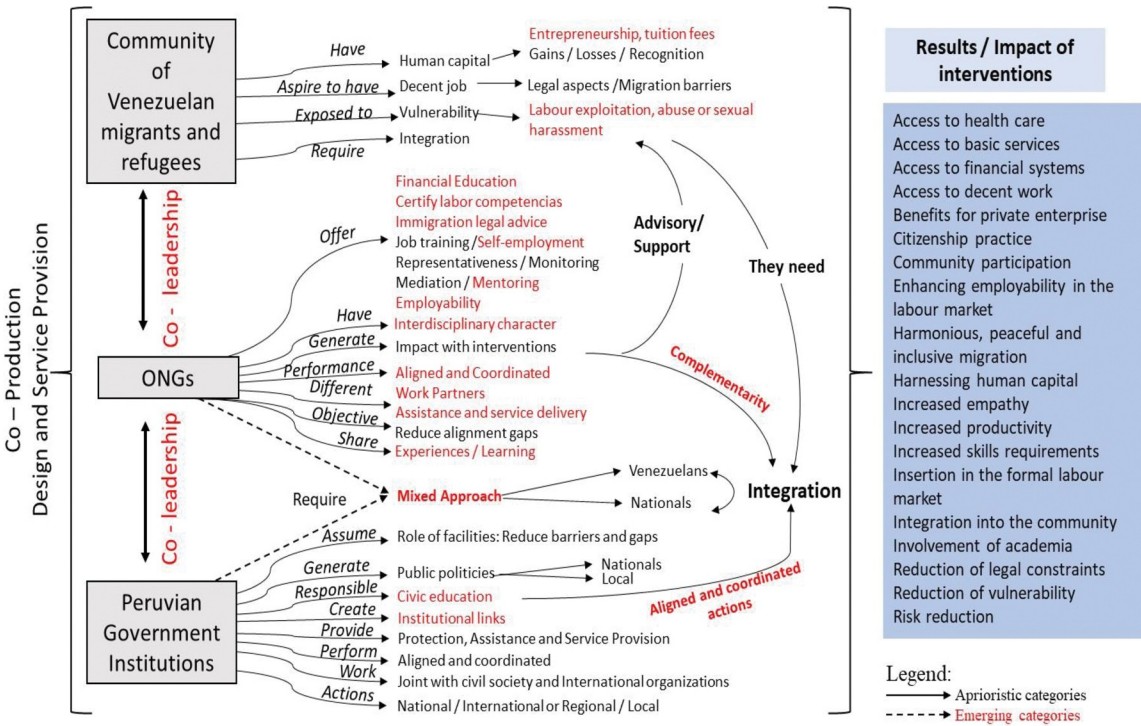

**Fig 1. Co-production in migration processes.**

and implementing policies at various levels, including national, regional, and local, which are interdisciplinary in nature and encourage collaboration between different organisations to develop high-impact solutions directly for the refugee and migrant population and indirectly for the local population. Public organisations are responsible for seeking solutions or minimising existing problems from a multidisciplinary perspective. Therefore, from the socioeconomic perspective, it is the responsibility of the State, through its representatives, to propose public policies that enable the integration of migrants and refugees, considering current legislation and migration policies–to allow the utilisation of the human capital present in the territory.

Therefore, the national migration policy needs to be regularly updated and adapted to the current context in the concerned territory, to increase the State's capacity to respond to the existing reality. In terms of migrant rights, while general aspects are addressed, special attention needs to be given to the regularization of migration, facilitation, and accessibility of information on legal procedures related to integration into the labour market under decent work conditions. Also, state responsibility extends to local governments, as the stance presented by municipalities can impact policy implementation practices, processes, communication, receptivity, and the attitude of citizens towards migrants and refugees. Moreover, it is also essential to acknowledge the role of public officials in organisational management, achieving results and managing processes, programmes, and projects–with standardised, regulated, and unbiased attention. Finally, the State plays also a vital role in supporting civil society organisations by complementing their activities, promoting their existence, and collaborating so they can act within an institutional framework.

**4.2.2. Role of private organisations.** In Peru, it is noteworthy that the private sector holds considerable economic and political influence. However, the current management practices in the Peruvian labour market reveal certain limitations regarding productivity

improvements, recognising the benefits of employing migrants and refugees, and addressing the issue of unfulfilled labour demand. Furthermore, there is a significant population of skilled professionals who are experiencing underemployment, either due to limitations in migration regularisation or difficulties in obtaining professional recognition through bureaucratic channels. Likewise, many employers seek to minimise costs by offering compensation below the market value and prioritising the hiring of vulnerable migrants and refugees under non-decent working conditions. This perpetuates a vicious circle that hinders the potential contribution of migration to sustainable development.

Therefore, the private sector plays a fundamental role in proposing solutions that enable the market to adjust, by leveraging the skilled workforce, presenting continuous training programs, and acquiring the necessary knowledge to generate economic growth and livelihood opportunities. With the contribution of companies, productivity can be enhanced through their collective expertise and capabilities. Moreover, in addition to fulfilling their corporate social responsibility, companies can increase their efficiency and impact by contributing to awareness and providing solutions for social issues. In this regard, it is relevant to develop effective communication strategies that include elements such as recognising the benefits of competitiveness, diversity, business loyalty, hiring individuals who face difficulties with job market insertion and increasing human capital by fostering experiences and skills. Additionally, at the internal management level, the challenge lies in recruiting, integrating, building relationships based on trust, security, openness, and well-being.

**4.2.3. Role of civil society organisations.**   Organised civil society, together with international organisations, seeks to complement the actions that the State fails to execute satisfactorily, thereby reducing gaps at various levels, such as information gap related to access to public services, exercising rights, other migration-related aspects, labour issues, and providing support for vulnerable groups. Organisations work to connect the efforts of various actors to avoid dismantling the impact of implemented actions and to broaden the scope of their impact.

Regarding the integration of migrants and refugees, the civil society organisations play a crucial role in providing services and support for training and technical assistance to help this population access job opportunities and represent citizens, defend rights, communicate demands to government entities, influence policy formulation, and generate in dialogue with different organisations. Also, civil society organisations can act directly by representing their constituencies in committees and commissions where relevant policies are discussed.

The relevance of international organisations and multilateral agencies, religious organisations and academia is also recognised. In general, it is confirmed that: 1. International organisations and multilateral agencies are responsible for transferring methodologies and capabilities to other involved organisations so that they can develop programmes; 2. As a result of foreign cooperation, financial contributions are also made to this subject matter—taking into account the limitations and income of the different territories; 3. From a more localised perspective, in community work, religious organisations act as points of encounter, care and guidance; 4. Religious organisations are relevant in enabling community strengthening to provide protection in places that are difficult to access or that require mechanisms to facilitate better communication flow; 5. Academia assumes the responsibility of analysing reality, identifying problems, presenting recommendations, supporting the reduction of obstacles, and facilitating the improvement of society's quality of life. Thus, each organisation has its role and relevance within the decision-making spiral, so that the public can achieve socioeconomic integration in the country. Therefore, working in alliance with different organisations makes it possible to increase prevention work or outreach to the target public, broadening the identification of the problem, territorial coverage, and communication potential.

*4.2.3.1 Role of migrant and refugee-based organisations.* Also, within the civil society organizations it is important to highlight the role of migrant and refugee-based organizations. The first level of contact with the migrant and refugee population in need of support occurs in grassroots organisations that are led by Venezuelans, who have better understanding of the reality and needs of the beneficiary public. The people, who are part of these grassroots organisations are members of the population themselves and can interpret the reality and the approaches of this population more accurately, leading to better communication among their fellow citizens. This allows the organisations to understand the needs of the migrant population, have easy access to them, establish links and disseminate information among nationals, ultimately bringing the migrant community, the population, and national organisations closer together.

The efforts of migrant-based NGOs are of relevance, as they can mobilise a network to facilitate access to resources and dialogue, understand social tensions existing between the host and migrant communities, and seek ways to generate cooperation and spaces for integration. Also, they have an important role in decision-making processes, strategy development and formulation of public policies. The participation of beneficiary representatives is crucial in offering and sharing a vision that allows for the identification of failures, inequities, or gaps that need to be addressed. Thus, migrant-based organisations represent, for many other organisations, a possibility of having a closer approach to the population with more efficiency, efficacy and effectiveness.

With this, the grassroots organisations provide economic support for development, education and training, as well as identifying vulnerable cases, organising humanitarian distributions and integration events. Consequently, these NGOs act as unofficial representatives, or "ambassadors" of the community, fulfilling advocacy roles and operating the initiatives sponsored by international and national agencies. So, gaps are reduced at various levels and needs associated with various vulnerabilities are addressed.

Fig 2 shows the role of the different organizations involved in the co-production of products and services delivered to migrants and refugees.

## 5. Discussion

The main barrier for integration of migrant and refugee human capital into the labour market in Metropolitan Lima, Peru, while ensuring decent work, lies in the legal aspect and regulatory —confirming Hinterleitner et al [41] proposition. Migrants are inserted into a highly informal labour structure with low productivity [43], and to effectively address the regulatory challenges in terms of migration and labour policies, it is necessary to understand their complexity, adapt it to the local context, facilitate access to relevant information, ensure the adequate preparation of public servants and adapt the corresponding bureaucratic procedures.

From the studied reality it is possible to confirm that, although possessing a legal status does necessarily guarantee improved employability [43, 45], migrants who do not have the corresponding documentation tend to be more exposed to various factors of vulnerability of their rights [9, 55], such as labour exploitation, harassment, or sexual harassment [3, 20]. Once the legal barrier has been overcome, a multidisciplinary nature of issues increases–such as basic needs, employability, access to the formal labour market and others—the scope of the demand of the target audience, which cannot be carried out by a single organisation alone.

Therefore, from the studied case, it is possible to verify that the co-production in mass migration processes involves an interdisciplinary effort aimed at supporting and addressing complementary needs of a public exposed to different vulnerabilities, which often compound each other. This effort is relevant, given that the support of different organisations makes it possible to generate more confidence and security for migrants–confirming the assertions of

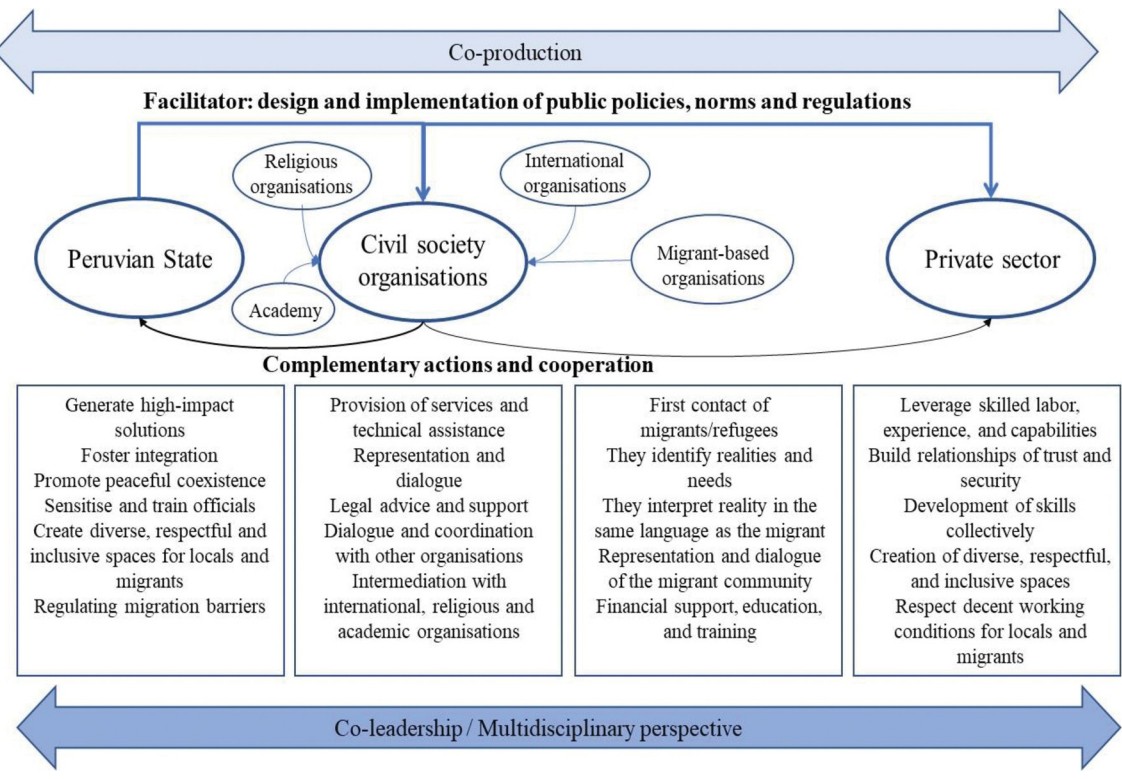

**Fig 2. The role of organisations in migration co-production processes.**

Fernandes, Prabawa, and Therik [74] and Osborne, Radnor, and Strokosch [60]. Then, it is necessary the creation of a coordination network to expand humanitarian assistance, integration efforts, and to contribute to territorial development. This approach represents a learning process for organisations that enables them to build institutional capacity by improving the quality of care provided and thereby amplifying their impact. Its effectiveness is contingent on continuous learning, identifying improvement opportunities, engaging with the public and prioritising vulnerable populations. Also, organisational initiatives facilitate reducing the gaps in care, in a multidisciplinary and complementary way, such that the efforts of various organisations can provide direct or indirect support for diverse challenges. Consequently, the actions offered by the various organisations expand their potential contribution to the host society, creating a type of positive externality.

Furthermore, as pointed out by Gilmartin and Dagg [70], the collaboration between migrants, non-governmental organizations and the State facilitates the discussion of challenges and solutions for integrating minorities into society. Albert et al. [59] and Osborne, Radnor and Strokosch [60] highlighted the active and participatory role the population plays in designing, developing, and delivering services.

Therefore, the co-production efforts, led by a co-leadership model [67, 68], such as those implemented by the Working Group on Refugees and Migrants, create spaces for coordinating actions, sharing information and experiences, expanding learning processes, enhancing cooperation and complementation practices, maximising the impact of actions at a lower cost and with greater efficiency to integrate migrants at a decent work reality–or a less unfair situation. Also, among these stands out co-leadership, as an expression of joint work between migrants and organisations–such as mentioned by Schlappa and Imani [67] and Gibeau et al. [68].

Co-production aims to improve the quality of services [60, 65, 69], which enables the expansion of installed capacity and the improvement of services´ quality through shared learning. As a result, it reduces gaps and increases the potential to serve diverse audiences, including nationals who may be in similar vulnerable situations. In the context of migration, human capital and decent work, co-production process directly and indirectly contributes to the creation of livelihoods and subsequent improved quality of life. This requires multidisciplinary and comprehensive actions that involve various resources.

Creating spaces that allow a closer approach to the beneficiary population can expand the potential for support by facilitating communication processes and identifying needs and improvement opportunities in care and support. Co-created strategies generate livelihoods and improve the quality of life in a sustainable manner over time. Proximity makes it possible to identify vulnerabilities and groups that require prioritisation. This comprehensive effort of complementation and cooperation involves diverse actions, emphasising the role of the State, private organisations, and civil society organisations.

It is up to the State to assume the role of facilitator, from different perspectives, creating and implementing multidisciplinary public policies at different levels and providing appropriate regulations for the context. In this regard, the host country needs to provide support in terms of governance and protection of migrant workers [3], which involves identifying migration barriers that limit the development of the skills of new human capital [41] and recognising its role in the integration process of new human capital [49] as well as in the "human capital gain" or "brain gain"–such as proposed by Murphy and Pacher [7]. Therefore, the State's responsibility transcends to different levels of government and involves recognising the role of public officers and providing them with appropriate training to execute processes, programmes and projects that are compatible with the existing reality.

Additionally, the private sector has an important role in contributing to social issues and allowing the market to adjust by leveraging human capital and increasing productivity. In that sense, it is necessary to work on communication, adapt management practices and build spaces aimed at creating diversity and avoiding the waste of human capital. It is relevant to highlight the significance of human capital to economic growth, if it is applied and used correctly [34]. Therefore, it is the responsibility of managers to raise awareness, and facilitate integration under decent work conditions that encourage creation of a fair and just context. Likewise, new human capital adaptation depends on the host culture, in creating positive relations between migrants and the local population, with the objective of ensuring the optimal development of their potential [35]. Conversely, the private sector can also perpetuate a circle of limiting the potential contribution of migration to sustainable development, particularly when prejudice affects access to higher-income jobs that migrants and refugees can opt for [44, 45], due to their migratory status and consequent vulnerability.

The civil society organisations are crucial in contributing to complement actions that the State is unable to fully address. Their attention often addresses gaps at various levels, reaching not only the migrant and refugee population, but also nationals who face similar vulnerable conditions, particularly in developing countries. Civil society organisations play a political role in representing their audiences, which involves advocating for their demands in decision-making processes to expand the potential for integration in the destination territory. It is important to note that the socioeconomic integration of immigrants is one of the fundamental aspects for achieving common prosperity, harmony, and social stability [40]. Through alliances and the creation of organisation networks, civil society organisations can maximize their impact in achieving these goals.

Last but not least, grassroots organisations provide what Gilmartin and Dagg [70] characterise as an environment where migrants feel represented and at the same time manage to

create influence in terms of governance and representativeness. These organisations allow sharing responsibilities and power in the production and delivery of services from the specialised perspective of the beneficiary public. By understanding the reality experienced by the beneficiary public, mobilising resources, understanding conflicts, and facilitating communication and information processes, these organisations are fundamental to increase the quality of care provided and make a positive impact on the integration process. Finally, the learning processes experienced by these organisations and their audiences become lessons, which lead to higher levels of maturity and efficiency in customer service. Then, these organisations assume a political role in intermediation and attention to the demands of their audiences, bringing the subjects into the political agenda. This active participation in the co-production process highlights their fundamental role in the development and delivery of public services [59, 60] and in the execution and implementation of quality and efficient public policies [60, 61].

In the context of limited resources, the formation of network and governance structure become fundamental. These networks, comprised of various stakeholders, enable the improvement of rights-based migration governance [72, 73] and address the need to provide immigrants with the necessary tools to integrate into the labour market, and contribute to the economic growth of the destination territory [48], under decent work conditions that provide productive, well-paid job opportunities with security and social protection for families [3, 50]. Therefore, the work of civil society organisations, particularly those of migrant base, is crucial in situations of crisis contexts and in developing countries where people become more susceptible to vulnerabilities. As a result, emphasising the importance of decent work involves enhancing the key areas highlighted by Gibb et al. [51], and Schulte et al. [52] in terms of justice, job creation, social protection, workers' rights, and social dialogue.

## 6. Conclusion

This study objective was to explain the co-production process of integrating Venezuelan migrants and refugees into the host society under decent work conditions. It allowed us to recognize that among the innumerable barriers that increase the vulnerabilities faced by migrants and refugees in the processes of integration into the host society, the primary hurdle is related to legal matters. Given the limited legal recognition, migrants and refugees find themselves in a condition that affects different aspects of their lives, such as their ability to accumulate livelihoods, improve quality of life and, therefore, contribute to the development of the territory in which they reside. Moreover, this reality worsens in territories where labour relations are characterised by informality, as in developing countries, since it allows the use of mechanisms related to violations and abuses of labour rights. Consequently, all the factors mentioned above restrict the possibility of obtaining decent work, and the co-production effort acts as a catalytic actor in generating a more favourable context to integrate migrants and create adequate working conditions.

This study represents a theoretical advancement by identifying that, in the specific case of mass migration processes and the consequent integration of human capital, the relevance of organised civil society organisations lies in the collective creation of mechanisms and spaces that facilitate communication and information exchange between the beneficiary public and the organisations involved. They do so by identifying existing problems and vulnerabilities and alternatives to mitigate them. Additionally, in processes of mass migration, migrant and refugee-based civil society organisations play a fundamental role, as they can enhance the quality of outreach, trust and dialogue with beneficiaries who lack rights, resources, and access to services. That is done by identifying specific needs and potential solutions and, therefore, improving the quality and efficiency of the services provided. The aim is not to absolve the

State or private organisations of their responsibilities, but rather to enable civil society organisations to make a meaningful contribution—towards attaining Sustainable Development Goals -, in contexts of high vulnerability and limited resources. In addition, the creation of governance networks, with the participation of civil society organisations and representatives of the beneficiary public, can facilitate the implementation of strategies that benefit both migrants and the host population in mass migration processes; respecting rights of all individuals and promoting the construction of fairer and more equitable societies.

In practical terms, real change depends to a large extent on the co-production process, which encompasses public, private, and organised civil society organisations. This effort involves a collective and interdisciplinary effort, utilising different perspectives, capabilities, resources, and cross-cutting activities, to generate change for the beneficiary public. The achievement of results depends on complementarity, participation, communication, coordination, organisation, and outreach work with the beneficiary public. Also, the process also involves identifying cultural traits between the host and migrant communities, so that the exchange can facilitate the creation of positive relationships. Therefore, the effort also involves recognising the needs of both the migrant and host populations, so that the process of insertion does not result in exclusion or differentiation between migrants, refugees, and the host population. Co-production then allows the generation of shared learning, expands installed capacity, and improves the quality of services, reducing gaps in care for migrants and generating positive externalities for the society. Integration efforts through co-production maximise the potential contribution of human capital to the host territory, increasing productivity and potential to contribute to sustainable territorial development. The aim is to move away from a vicious cycle of human capital waste towards a virtuous cycle of human capital gain through fair practices and decent employment opportunities. Therefore, it is the responsibility of different organisations to identify their potential for action and the possibilities for interaction between the different actors.

Finally, this study has identified opportunities for future research that specifically consider the role of migrant-based civil society organisations in processes of integration in mass migration processes. It also identifies the relevance of studies on co-production efforts targeted towards audiences in conditions of increased vulnerability due to gender, age, or health issues. Among its limitations we may highlight that it represents a case, under a specific context, that may not be completely applied under different realities.

## Author Contributions

**Conceptualization:** Valentina Gomes Haensel Schmitt, Agnieszka Ewa Olter-Castillo, Mirza Marvel Cequea.

**Data curation:** Valentina Gomes Haensel Schmitt, Agnieszka Ewa Olter-Castillo, Mirza Marvel Cequea.

**Formal analysis:** Valentina Gomes Haensel Schmitt, Agnieszka Ewa Olter-Castillo, Mirza Marvel Cequea.

**Funding acquisition:** Valentina Gomes Haensel Schmitt, Mirza Marvel Cequea.

**Investigation:** Valentina Gomes Haensel Schmitt, Agnieszka Ewa Olter-Castillo, Mirza Marvel Cequea, Helder Huaranga Chayña.

**Methodology:** Valentina Gomes Haensel Schmitt, Agnieszka Ewa Olter-Castillo, Mirza Marvel Cequea.

**Project administration:** Valentina Gomes Haensel Schmitt, Mirza Marvel Cequea.

**Resources:** Valentina Gomes Haensel Schmitt.

**Supervision:** Valentina Gomes Haensel Schmitt, Mirza Marvel Cequea.

**Validation:** Valentina Gomes Haensel Schmitt, Agnieszka Ewa Olter-Castillo, Mirza Marvel Cequea.

**Visualization:** Valentina Gomes Haensel Schmitt, Agnieszka Ewa Olter-Castillo, Mirza Marvel Cequea, Helder Huaranga Chayña.

**Writing – original draft:** Valentina Gomes Haensel Schmitt, Agnieszka Ewa Olter-Castillo, Mirza Marvel Cequea, Helder Huaranga Chayña.

**Writing – review & editing:** Valentina Gomes Haensel Schmitt, Agnieszka Ewa Olter-Castillo, Mirza Marvel Cequea, Helder Huaranga Chayña.

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
