## [Decision Letter · Decision Letter 0]

27 Sep 2023

PONE-D-23-26147Co-Production for the integration of migrant human capital into the decent workPLOS ONE

Dear Dr. Gomes,

Thank you for submitting your manuscript to PLOS ONE. After careful consideration, we feel that it has merit but does not fully meet PLOS ONE’s publication criteria as it currently stands. Therefore, we invite you to submit a revised version of the manuscript that addresses the points raised during the review process. 

We look forward to receiving your revised manuscript.

Kind regards,

Cesar Infante Xibille, Ph.D

Academic Editor

PLOS ONE

2. Please provide additional details regarding participant consent. In the Methods section, please ensure that you have specified (1) whether consent was informed and (2) what type you obtained (for instance, written or verbal).

Reviewers' comments:

Reviewer's Responses to Questions

**Comments to the Author**

1. Is the manuscript technically sound, and do the data support the conclusions?

Reviewer #1: Yes

2. Has the statistical analysis been performed appropriately and rigorously? 

Reviewer #1: N/A

3. Have the authors made all data underlying the findings in their manuscript fully available?

Reviewer #1: Yes

4. Is the manuscript presented in an intelligible fashion and written in standard English?

Reviewer #1: Yes

5. Review Comments to the Author

Reviewer #1: It is an interesting exploratory study on labour migration in Latin America. Overall, a sound research design, and well executed analysis. MU main problem concerns the conceptual framework on which the study is based.

Literature review presents a stunted, and arbitrarily selected angle in the presented maniuscript as it argues that “migration movements can have a positive impact on social and economic welfare of the host country, as they can rejuvenate the working population and intensify the labour force.” (Lines 91-92).

The “can” suggests that it cannot, but doesn’t have to and this raises the question of short- versus long-term consequences as well as proportionality. Being a migrant myself, I will be the last one to argue that migration does not have a positive impact on human potential but to claim that it is solely or overwhelmingly positive, would be too erroneous.

The claimed economic benefits of migrationare wildly overstated in the conceptual framework. The authors ignore or understate crowding and congestion effects as well as the importance of positional goods (amenity/status goods rationed in supply such that they gain value from others not having them). Most importantly, the manuscript obscures the downward pressures immigration puts on wages, suppressing wages rather than cutting them. The Baumol Effect means that competition for high productivity workers raises wages in occupations that do not enjoy increases in productivity. Using migration to make labour more plentiful compared to capital suppresses the Baumol effect and so suppresses wages. Hence the pandemic pause on migration led to (money) wages at the lower end of  labour markets starting to rise noticeably in the US and the UK: the Baumol Effect was no longer being suppressed. As Per Peter Turchin, it is a not a coincidence that US growth in real wages essentially stopped a few years after mass migration to the US resumed. Canada is another case showcasing this dynamic very clearly at the moment. It is hard to believe that Latin America is an outlier case. This is a dimension that should be addressed in revisions. Two pointers where to start below, but I am confident the authors will be able to find literature on the subject without any problems.

George Borjas, ‘Immigration and the American Worker: A Review of the Academic Literature,’ Center for Immigration Studies, April 2013.

David Card, Christian Dustmann and Ian Preston, ‘Immigration, Wages, and Compositional Amenities,’ Norface Migration Discussion Paper No. 2012-13, February 2012.

6. PLOS authors have the option to publish the peer review history of their article (what does this mean?). If published, this will include your full peer review and any attached files.

Reviewer #1: No

---

## [Author Response · Author response to Decision Letter 0]

8 Nov 2023

COMMENT 1. Is the manuscript technically sound, and do the data support the conclusions?

Reviewer #1: Yes

AUTHORS’ RESPONSE: We would like to thank you for taking the time to review thoroughly our material. We have considered all your recommendations and added them to this recent version. We hope it now meets all your expectation.

COMMENT 2. Has the statistical analysis been performed appropriately and rigorously?

Reviewer #1: N/A

COMMENT 3. Have the authors made all data underlying the findings in their manuscript fully available?

Reviewer #1: Yes

AUTHORS’ RESPONSE: Thank you for recognizing it.

COMMENT 4. Is the manuscript presented in an intelligible fashion and written in standard English?

Reviewer #1: Yes

AUTHORS’ RESPONSE: Thank you for recognizing it.

COMMENT 5. Review Comments to the Author

Reviewer #1: It is an interesting exploratory study on labour migration in Latin America. Overall, a sound research design, and well executed analysis. MU main problem concerns the conceptual framework on which the study is based.

Literature review presents a stunted, and arbitrarily selected angle in the presented maniuscript as it argues that “migration movements can have a positive impact on social and economic welfare of the host country, as they can rejuvenate the working population and intensify the labour force.” (Lines 91-93).

The “can” suggests that it cannot, but doesn’t have to and this raises the question of short- versus long-term consequences as well as proportionality. Being a migrant myself, I will be the last one to argue that migration does not have a positive impact on human potential but to claim that it is solely or overwhelmingly positive, would be too erroneous.

The claimed economic benefits of migrationare wildly overstated in the conceptual framework. The authors ignore or understate crowding and congestion effects as well as the importance of positional goods (amenity/status goods rationed in supply such that they gain value from others not having them). Most importantly, the manuscript obscures the downward pressures immigration puts on wages, suppressing wages rather than cutting them. The Baumol Effect means that competition for high productivity workers raises wages in occupations that do not enjoy increases in productivity. Using migration to make labour more plentiful compared to capital suppresses the Baumol effect and so suppresses wages. Hence the pandemic pause on migration led to (money) wages at the lower end of labour markets starting to rise noticeably in the US and the UK: the Baumol Effect was no longer being suppressed. As Per Peter Turchin, it is a not a coincidence that US growth in real wages essentially stopped a few years after mass migration to the US resumed. Canada is another case showcasing this dynamic very clearly at the moment. It is hard to believe that Latin America is an outlier case. This is a dimension that should be addressed in revisions. Two pointers where to start below, but I am confident the authors will be able to find literature on the subject without any problems.

George Borjas, ‘Immigration and the American Worker: A Review of the Academic Literature,’ Center for Immigration Studies, April 2013.

David Card, Christian Dustmann and Ian Preston, ‘Immigration, Wages, and Compositional Amenities,’ Norface Migration Discussion Paper No. 2012-13, February 2012.

AUTHORS’ RESPONSE:

Upon careful consideration, we have chosen to replace the term “can” with “might,” thereby focusing on the potential positive implications of migration. However, to maintain a balanced perspective and acknowledge the potential challenges associated with migration, we have incorporated insights drawn from the esteemed works of researchers such as David Card and Joseph G. Altonji. Their studies conclude that, at the local level, immigration wields influence over a range of factors, encompassing wages, tax structures, demographic composition, and labor market dynamics. These dynamics may engender competitive elements between migrants and native residents, potentially unsettling the socio-demographic equilibrium that native inhabitants typically encounter in their daily experiences.

The previous version:

In migration processes, the adaptation of new human capital is intended to foster positive relations between migrants and local communities with the aim of ensuring the optimal development and utilisation of its potential [35]. Integration can be understood as a ¨two-way¨ process of mutual accommodation of host communities and immigrants [36]. Socioeconomic integration of immigrants is considered one of the most important paths when seeking common prosperity, harmony, and social stability [37]. In this sense, migration movements can have a positive impact on social and economic welfare of the host country, as they can rejuvenate the working population and intensify the labour force. Therefore, it is essential that the host country provides support in terms of governance and protection of migrant workers [3].

The modified version (lines 71-85:

“In migration processes, the adaptation of new human capital is intended to foster positive relations between migrants and local communities with the aim of ensuring the optimal development and utilisation of its potential [35]. At the local level, immigration not only impacts wages and taxes but also alters the composition of the local population [36] and influences the labour market dynamics [37]. This shift may potentially disrupt the socio-demographic balance that native inhabitants experience in their day-to-day lives [36, 37] and possibly raise more concerns among individuals with lower educational attainment [36]. Moreover, the migration of individuals with lower skill levels may lead to modest competition between migrants and native inhabitants [38] . Integration can be understood as a ¨two-way¨ process of mutual accommodation of host communities and immigrants [39]. Socioeconomic integration of immigrants is considered one of the most important paths when seeking common prosperity, harmony, and social stability [40]. In this sense, migration movements might have a positive impact on social and economic welfare of the host country, as they can rejuvenate the working population and intensify the labour force. Therefore, it is essential that the host country provides support in terms of governance and protection of migrant workers [3]”

COMMENT 6. PLOS authors have the option to publish the peer review history of their article (what does this mean?). If published, this will include your full peer review and any attached files.

Do you want your identity to be public for this peer review? 

Reviewer #1: No

---

## [Decision Letter · Decision Letter 1]

29 Nov 2023

Co-Production for the integration of migrant human capital into the decent work

PONE-D-23-26147R1

Dear Dr. Gomes Hansel

We’re pleased to inform you that your manuscript has been judged scientifically suitable for publication and will be formally accepted for publication once it meets all outstanding technical requirements.

Kind regards,

Cesar Infante Xibille, Ph.D

Academic Editor

PLOS ONE

Additional Editor Comments (optional):

Reviewers' comments:

Reviewer's Responses to Questions

**Comments to the Author**

1. If the authors have adequately addressed your comments raised in a previous round of review and you feel that this manuscript is now acceptable for publication, you may indicate that here to bypass the “Comments to the Author” section, enter your conflict of interest statement in the “Confidential to Editor” section, and submit your "Accept" recommendation.

Reviewer #1: All comments have been addressed

2. Is the manuscript technically sound, and do the data support the conclusions?

Reviewer #1: Yes

3. Has the statistical analysis been performed appropriately and rigorously? 

Reviewer #1: N/A

4. Have the authors made all data underlying the findings in their manuscript fully available?

Reviewer #1: Yes

5. Is the manuscript presented in an intelligible fashion and written in standard English?

Reviewer #1: Yes

6. Review Comments to the Author

Reviewer #1: I am pleased to note that the revised version of the manuscript has successfully addressed all the comments I provided. The thorough attention given to the feedback demonstrates a commendable commitment to enhancing the quality of the work. The revisions have significantly strengthened the manuscript, and I appreciate the author's dedication to incorporating the suggestions. The clarity of presentation and the thoughtful responses to the points raised during the review process have notably improved the overall conceptual coherence and quality of the paper.

7. PLOS authors have the option to publish the peer review history of their article (what does this mean?). If published, this will include your full peer review and any attached files.

Reviewer #1: No

---

## [Editor Report · Acceptance letter]

2 Dec 2023

PONE-D-23-26147R1 

Co-Production for the integration of migrant human capital into the decent work 

Dear Dr. Schmitt:

I'm pleased to inform you that your manuscript has been deemed suitable for publication in PLOS ONE. Congratulations! Your manuscript is now with our production department. 

Kind regards, 

on behalf of

Dr. Cesar Infante Xibille 

Academic Editor

PLOS ONE